# Peroxisome Proliferator-Activated Receptor α Attenuates Hypertensive Vascular Remodeling by Protecting Vascular Smooth Muscle Cells from Angiotensin II-Induced ROS Production

**DOI:** 10.3390/antiox11122378

**Published:** 2022-11-30

**Authors:** Ye Liu, Yan Duan, Nan Zhao, Xinxin Zhu, Xiaoting Yu, Shiyu Jiao, Yanting Song, Li Shi, Yutao Ma, Xia Wang, Baoqi Yu, Aijuan Qu

**Affiliations:** 1Department of Physiology and Pathophysiology, School of Basic Medical Sciences, Capital Medical University, Beijing 100069, China; 2Key Laboratory of Remodeling-Related Cardiovascular Diseases, Ministry of Education, Beijing 100069, China; 3Department of Cardiology, Cardiovascular Key Laboratory of Zhejiang Province, The Second Affiliated Hospital, College of Medicine, Zhejiang University, Hangzhou 310009, China; 4Department of Pathology, Beijing Anzhen Hospital, Capital Medical University, Beijing 100029, China

**Keywords:** vascular remodeling, oxidative stress, vascular smooth muscle cell, peroxisome proliferator-activated receptor α

## Abstract

Vascular remodeling is the fundamental basis for hypertensive disease, in which vascular smooth muscle cell (VSMC) dysfunction plays an essential role. Previous studies suggest that the activation of peroxisome proliferator-activated receptor α (PPARα) by fibrate drugs has cardiovascular benefits independent of the lipid-lowering effects. However, the underlying mechanism remains incompletely understood. This study explored the role of PPARα in angiotensin II (Ang II)-induced vascular remodeling and hypertension using VSMC-specific *Ppara*-deficient mice. The PPARα expression was markedly downregulated in the VSMCs upon Ang II treatment. A PPARα deficiency in the VSMC significantly aggravated the Ang II-induced hypertension and vascular stiffness, with little influence on the cardiac function. The morphological analyses demonstrated that VSMC-specific *Ppara*-deficient mice exhibited an aggravated vascular remodeling and oxidative stress. In vitro, a PPARα deficiency dramatically increased the production of mitochondrial reactive oxidative species (ROS) in Ang II-treated primary VSMCs. Finally, the PPARα activation by Wy14643 improved the Ang II-induced ROS production and vascular remodeling in a VSMC PPARα-dependent manner. Taken together, these data suggest that PPARα plays a critical protective role in Ang II-induced hypertension via attenuating ROS production in VSMCs, thus providing a potential therapeutic target for hypertensive diseases.

## 1. Introduction

Hypertension is one of the most prevalent cardiovascular diseases [1]. Long-term high blood pressure can thicken and stiffen the artery walls [2]. The core pathological alterations of hypertension are decreased compliance and increased stiffness, which can eventually lead to significant cardiac, cerebral, and renal problems and are the main risk factors for stroke and coronary heart disease [3].

The major pathogenic mechanism in a succession of hypertension problems is vascular remodeling [4]. The primary affecting ingredient within the renin-angiotensin system (RAS), angiotensin II (Ang II), has a significant impact on blood pressure regulation, hypertension development, and vascular remodeling, as well as the production of vascular inflammation and oxidative stress-inducing processes [5]. Endothelial cells and vascular smooth muscle cells (VSMCs) mediate the aforementioned actions in resident vascular cells [6]. The main cellular components in the arteries are vascular smooth muscle cells (VSMCs), which have activities that keep the vessels’ structural and physiological integrity [7]. Contraction and blood pressure control are the primary tasks of VSMCs. VSMCs become highly proliferative in hypertension and produce a high level of extracellular matrix components, such as collagen and elastin, which all contribute to vascular remodeling and stiffness [8]. Hemodynamic, reactive oxygen species (ROS), and vasoactive chemicals such as Ang II and aldosterone have all been shown to affect these alterations [9]. Mitochondria are both the target and the supplier of reactive oxygen species (ROS). The overload of oxidant radicals’ damages mitochondria and causes a malfunction. Damaged mitochondria are eliminated via quality control mechanisms such as mitochondrial dynamics, mitophagy, and mitochondrial biogenesis to preserve homeostasis [10]. In proliferative VSMCs, a mitochondria transition from fusion to fission in response to oxidative stress, making them become tiny and disordered [11,12]. Mitophagy and mitochondrial biogenesis can be pharmacologically activated to alleviate the mitochondrial dysfunction, oxidative metabolism, and cardiovascular disease [13].

The peroxisome proliferator-activated receptor α (PPARα) is a member of the PPAR nuclear receptor subfamily [14] that heterodimerizes with the retinoid X receptor (RXR) to activate transcription by binding to the PPAR response element (PPRE), also known as direct repeat 1(DR1). The PPRE/DR1 consensus sequence is AGGTCANAGGTCA, which consists of two AGGTCA sequences separated by one nucleotide, with PPARα and RXR binding to the 5′ and 3′ AGGTCA sequences, respectively [15]. The upstream steps of the FA catabolic pathway, including FA uptake, transport, and oxidation, intervene in PPAR/RXR target genes with a functioning PPRE/DR1 [16]. PPARα has been identified in prior findings to lessen the risk of cardiovascular disease by upregulating the genes associated with the glucose and lipid metabolism [17]. The majority of studies found that PPARα agonists protect the cardiovascular system both in basic research and in clinical applications [18]. PPARα is involved in VSMC apoptosis, phenotypic transition, and proliferation [19,20]. However, the impact of VSMC PPARα on vascular injury remains unclear.

Mice with a specific PPARα disruption in VSMCs were used to investigate the effect of VSMC PPARα in Ang II-induced vascular remodeling and the underlying mechanisms. The knockdown of VSMC PPARα significantly aggravated the Ang II-induced hypertension and vascular stiffness. The mechanism is mainly the increased production of mitochondrial reactive oxidative species (ROS) in Ang II-treated primary VSMCs, a process alleviated by the activation of PPARα by Wy14643. This study provides potential targets for the prevention and treatment of hypertensive diseases.

## 2. Materials and Methods

### 2.1. Animal Experiments

All animal care and experimental procedures were approved under a project license (AEEI-2018-127) granted by the ethics board of Capital Medical University. *Ppara*^fl/fl^ mice on a C57BL/6J background were generated as previously described [21]. *Ppara*^fl/fl^ mice were crossed with SM22α-Cre to produce VSMC-specific *Ppara*-deficient mice (*Ppara*^ΔSMC^) [22]. Two-month-old *Ppara*^ΔSMC^ and *Ppara*^fl/fl^ male mice were infused with saline or Ang II at 490 ng/kg per minute with micro-osmotic pumps (Alzet, Model 1004, Cupertino, CA, USA) for 28 days as described [23]. Their blood pressure was monitored by the tail–cuff method.

### 2.2. WY14643 Treatment

To examine whether Wy14643 attenuated Ang II-induced vascular remodeling and hypertension in a VSMC PPARα-dependent manner, *Ppara*^fl/fl^ and *Ppara*^ΔSMC^ male mice with a saline or Ang II infusion were administered a chow or WY14643 (A4305, APExBIO, Houston, TX, USA) diet (0.1% in chow diet) for two weeks [24]. The composition of the chow diet is listed in Appendix A.

### 2.3. Measurement of Aortic Stiffness and Cardiac Function

Aortic stiffness in the left common carotid artery was assessed by measuring the pulse wave velocity (PWV), pulse pressure variation (PPV), and cardiac function in vivo using a Vevo 2100 micro-ultrasound imaging system (FUJIFILM VisualSonics, Bothell, WA, USA). For the PWV measurements, *Ppara*^fl/fl^ and *Ppara*^ΔSMC^ mice were anesthetized by isoflurane inhalation. The transit-time method estimated the PWV from the pressure wave transit time between the two measurement locations separated by a known distance, as previously described [25]. Briefly, the PWV was determined by the simultaneous tracking of the R-wave of the ECG and the pulse wave along with the 2 locations of suprarenal left common carotid aorta. Vevo 2100 software was used to calculate the PWV as a ratio of the distance (d) between the 2 locations along the aorta and the time delay (Δt) of the pulse wave between these locations and is expressed in m/s.

For the cardiac function assessment, the cardiac size, shape, and function were analyzed by conventional two-dimensional imaging and M-Mode recordings. The left ventricular anterior wall (LVAW) thickness, the left ventricular internal diameter (LVID), and the left ventricular posterior wall (LVPW) thickness in both the systole and diastole were measured in short-axis M-mode images. The functional data were derived from both parasternal long-axis two-dimensional real-time 3-beat loops and M-mode measurements, including fractional shortening (FS%) and ejection fraction (EF%). All the values were based on the average of at least five measurements.

### 2.4. Histological Analysis

Mice were sacrificed by CO_2_ inhalation and perfused with saline. Aortas from *Ppara*^fl/fl^ and *Ppara*^ΔSMC^ mice were fixed in phosphate-buffered 4% formalin for 24 h and then embedded in a Tissue-Tek O.C.T. compound (4583, Sakura Finetek USA, Torrance, CA, USA). Frozen sections (7 μm) of the thoracic aorta from *Ppara*^fl/fl^ and *Ppara*^ΔSMC^ mice were continuously collected from the proximal to the distal. Hematoxylin (RY-ICH001a, Roby, Beijing, China) and eosin (RY-ICH002a, Roby, Beijing, China) (H&E) staining, elastin staining by Weigher’s elastic staining method (G1592, Solarbio, Beijing, China), and Masson’s trichrome staining were routinely performed as previously described [26]. The images were obtained using an Olympus laser scanning microscope (U-LH100-3, Olympus Corporation, Tokyo, Japan).

### 2.5. Immunohistochemistry Staining

The frozen sections were incubated with 3% (*v/v*) H_2_O_2_ for 15 min to quench the endogenous peroxidase activity and blocked with 10% goat serum (ZLI-9021, ORIGINE, Beijing, China) for 1 h at room temperature. The sections were incubated with anti-PPARα (1:100 dilution, PA1-822A, Invitrogen, Waltham, MA, USA) antibodies overnight at 4 °C. The biotinylated secondary antibodies of the enzyme-labeled goat anti-mouse/rabbit IgG polymer (GK600505-B, Gene Tech, Shanghai, China) were added, followed by an incubation at 37 °C for 1 h. The slides were then developed with a DAB chromogenic solution (GK600505, Gene Tech, Shanghai, China) and counterstained with hematoxylin. The images were acquired using an Olympus laser scanning microscope (U-LH100-3), and the intensity of the immunostaining was analyzed by the average of 5 slides with 5 random fields at ×40 per slide using Image J software (NIH, Bethesda, MD, USA) [26].

### 2.6. Immunofluorescence Staining

The frozen sections were permeabilized with 0.1% Triton X-100 (9002-93-1, Sigma, MA, UK) and blocked with 10% goat serum (ZLI-9021, ORIGINE, Beijing, China) for 30 min at room temperature. The sections were incubated with the following diluted primary antibodies overnight at 4 °C: PPARα (1:100 dilution, PA1-822A, Invitrogen, Waltham, MA, USA) and αSMA (1:200 dilution, ab2961, Abcam, Madison, WI, USA). The fluorescent secondary antibodies, goat anti-rabbit TRITC (ZF-0316, ORIGINE, Beijing, China) and goat anti-mouse FITC (ZF-0312, ORIGINE, Beijing, China), were mixed and added, and the slides were further incubated at 37 °C for 1 h. The sections were then incubated with 4′,6′-diamidino-2-phenylindole (DAPI) and mounted. The images were acquired using an Olympus laser scanning microscope (U-LH100-3, Olympus Corporation, TKY, JPN).

### 2.7. Dihydroethidium (DHE) Staining

To investigate the ROS production in vivo, a drop of PBS was added to each frozen aortic tissue (7 μm), which was first incubated at 37 °C for 30 min, then with 0.1 % Triton X-100 at 37 °C for 10 min, and subsequently with DHE (5 μmol/L in PBS) at 25 °C for 30 min in the dark. After washing with PBS, the sections were incubated with 4′,6′-diamidino-2-phenylindole (DAPI) and mounted. The images were acquired using an Olympus laser scanning microscope (BX53, Olympus Corporation, Tokyo, Japan).

### 2.8. Quantitative Real-Time PCR Analysis

The total RNA was extracted from the tissues and cells using a Trizol reagent (Ambion. Carlsbad, CA, USA). A total of 1 μg of total RNA was reversely transcripted into cDNA using the GoScript^TM^ Reverse Transcription system (Promega, Madison, WI, USA) following the manufacturer’s protocol. A quantitative real-time PCR (qPCR) was performed with an iCycler IQ system (Bio-Rad Hercules, CA, USA) using SYBR Green JumpStart Taq Ready Mix (TaKaRa, Shiga, Japan). The relative gene levels were calculated by the 2^−ΔΔCT^ method. *Actb* mRNA was used as a control. The primers for the qPCR assays are listed in Appendix A.

### 2.9. Primary VSMC Culture

The primary VSMCs were isolated from 2-month-old *Ppara*^fl/fl^ and *Ppara*^ΔSMC^ mice and maintained in a smooth muscle cell medium (SMCM) supplemented with 1% penicillin/streptomycin, 10% smooth muscle cell growth supplement (SMCGS), and 10% fetal bovine serum (FBS) at 37 °C in a humidified atmosphere of 5% CO_2_ and 95% air, as previously described [26]. Passages 3 to 7 of the VSMCs at a 70–80% confluence were used for the experiments. For the H_2_O_2_ stimulation, the VSMCs were starved for 12 h before treatment with 100 μM of H_2_O_2_ (Zhiyuan, Tianjin, China) for 48 h [27].

### 2.10. Measurement of Mitochondrial Membrane Potential

The mitochondrial membrane potential was measured by a mitochondrial membrane potential assay kit with JC-1 (C2006, Beyotime, China) according to manufacturer’s instruction. Briefly, 1 × 10^5^ VSMCs from *Ppara*^ΔSMC^ and *Ppara*^fl/fl^ mice upon the Ang II treatment (1 μmol/L) for 24 h were incubated with a 2 μM JC-1 probe in the dark for 20 min. The mitochondrial membrane potential was calculated as the ratio of the red to green fluorescence intensity according to the instructions.

### 2.11. MitoSOX Staining

Mitochondrial superoxide, an important source of reactive oxygen species (ROS) in VSMCs, was measured by staining with MitoSOX Red (Invitrogen, M36008, Waltham, MA, USA). The VSMCs (1 × 10^5^) from *Ppara*^ΔSMC^ and *Ppara*^fl/fl^ mice upon the Ang II treatment (1 μmol/L) for 24 h were loaded with MitoSOX Red (5 μM) for 10 min at 37 °C and then washed. The images were acquired using an Olympus laser scanning microscope (U-LH100-3, Olympus Corporation, Tokyo, Japan).

### 2.12. Statistical Analysis

All the statistical analyses were performed with GraphPad Prism version 8.0 (GraphPad Software, Inc., San Diego, CA, USA). The results are expressed as the mean ± standard error (SEM). For the normal distribution data, an unpaired Student’s *t*-test was used between the two groups. For the comparison of multiple groups, a one-way ANOVA was performed followed by Tukey multiple comparison analysis. *p* values < 0.05 were considered to be statistically significant for all of the tests.

## 3. Results

### 3.1. PPARα Was Decreased in the Vessel upon Ang II Treatment

To investigate whether PPARα is involved in Ang II-induced hypertension, the mRNA and protein levels of PPARα were measured in the aortas from Ang II-infused (490 ng/kg/min for 28 days) mice. qPCR analysis showed that the *Ppara* mRNA was significantly lower in the aortas from Ang II-infused mice than that in saline-infused controls (Figure 1A). Immunohistochemical staining demonstrated that the PPARα protein levels were reduced in the aortic media of the Ang II group, consistent with the changes in the *Ppara* mRNA levels (Figure 1B,C). Immunofluorescence staining revealed that the colocalization of PPARα with the VSMC marker α-SMA was dramatically decreased in Ang II-infused aortas (Figure 1D). These results revealed that PPARα was downregulated in the VSMCs upon an Ang II infusion, indicating the possible involvement of the VSMC PPARα in Ang II-induced hypertension.

### 3.2. PPARα Deficiency in VSMCs Aggravated Ang II-Induced Hypertension and Vascular Remodeling

To gain an insight into the function of the VSMC PPARα in Ang II-induced hypertension, VSMC-specific PPARα-deficient (*Ppara*^ΔSMC^) mice and their littermate controls (*Ppara*^fl/fl^) were used. The metabolic general parameters of *Ppara*^ΔSMC^ and *Ppara*^fl/fl^ mice had no significance (Appendix A). The Ang II infusion significantly increased both the systolic and diastolic pressure in *Ppara*^fl/fl^ mice, which was markedly augmented in *Ppara*^ΔSMC^ mice (Figure 2A). The vascular stiffness analysis by the PWV measurement showed that there was an increase in its basal level in *Ppara*^ΔSMC^ mice, which was magnified by the Ang II treatment compared with that in *Ppara*^fl/fl^ mice (Figure 2B). These results indicated that a PPARα deficiency in VSMCs was involved in Ang II-induced hypertension and vascular stiffness. To assess whether the cardiac dysfunction contributes to PPARα deficiency-exacerbated hypertension, the cardiac function was evaluated. However, no difference in the cardiac EF and FS were found between *Ppara*^ΔSMC^ and *Ppara*^fl/fl^ mice treated with Ang II (Figure 2C,D). Similarly, no difference was observed between *Ppara*^ΔSMC^ and *Ppara*^fl/fl^ mice in the key hemodynamic parameters, including the LVAW, LVID, and LVPW thickness (Figure 2E,F). The above results confirmed that a PPARα deficiency in VSMCs exacerbates Ang II-induced hypertension independent of cardiac dysfunction.

### 3.3. PPARα Deficiency in VSMCs Aggravated Ang II-Induced Vascular Remodeling

Vascular remodeling is the fundamental pathophysiology of hypertension [28]. To identify whether VSMC PPARα regulates vascular remodeling, a morphological analysis was performed. H&E staining demonstrated that the aortic wall thickness was significantly increased in *Ppara*^ΔSMC^ mice compared with *Ppara*^fl/fl^ mice upon an Ang II treatment (Figure 3A). Masson’s trichrome staining revealed that a PPARα deficiency in VSMCs greatly aggravated the collagen deposition in the media and the adventitia of the aortic tissues (Figure 3B). Weigert’s elastin staining showed no difference in the elastic fiber structure between the four groups (Figure 3C). These data suggested that a PPARα deficiency in VSMCs aggravated an Ang II-induced vascular remodeling.

### 3.4. PPARα Deficiency in VSMCs Aggravated Ang II-Induced Vascular ROS Production

Numerous studies indicate that the production of reactive oxygen species (ROS) and the consequent oxidative stress play an important role in Ang II-induced vascular injury and remodeling [29]. As indicated by DHE staining, an Ang II infusion induced an obvious ROS production in the vessel wall, which was significantly aggravated in *Ppara*-deficient mice (41.8 ± 3.8% in *Ppara*^ΔSMC^ + Ang II group versus 23.7 ± 4.5% in *Ppara*^fl/fl^ + Ang II group) (Figure 4A). Similarly, in VSMCs, a PPARα deficiency aggravated the Ang II-induced vascular ROS production (Figure 4B). The NADPH oxidase (NOX) family is a major source of vascular oxidative stress and NOX4 is the main isoform expressed in the VSMCs of large arteries [23]. Our further investigation showed that Ang II-induced the vascular elevation of NOX4 in *Ppara*^fl/fl^ mice. However, this effect was slightly increased by the PPARα deletion in *Ppara*^ΔSMC^ mice (Appendix A). NOX4 is recognized as an oncoprotein localized to mitochondria [30]. The expression of NOX4 has been shown to be induced in VSMC mitochondria and vasculature and is correlated with an increased aortic stiffening and atherosclerosis in aged mice [31]. Mitochondria represents one of the major sources of a cellular ROS generation [32]. To determine the effects of a PPARα deficiency on the VSMC mitochondrial function, the mitochondrial membrane potential (MMP) with JC-1 was detected. As indicated by decreased JC-1 aggregates but increased JC-1 monomers, the MMP was significantly lowered in the VSMCs exposed to Ang II, which was further decreased in PPARα-deficient VSMCs (Figure 4C). Moreover, the mitochondrial ROS production was also measured by MitoSOX. Ang II induced a marked mitochondrial ROS production in *Ppara*^fl/fl^ VSMCs, which was dramatically augmented in the *Ppara*-deficient VSMCs (Figure 4D). All the above data suggest that a PPARα deficiency aggravates the Ang II-induced vascular ROS production, indicating that PPARα has a protective role in vascular remodeling.

### 3.5. Wy14643 Attenuated Ang II-Induced Hypertension and Vascular Stiffness Mainly via VSMC PPARα-Dependent Mechanisms

Previous studies showed that PPARα agonist may exhibit cardiovascular benefits independent of its lipid-lowering effects. However, whether VSMC PPARα is involved in this process is unclear. WY14643 (4-chloro-6-(2,3-xylidino)-2-pyrimidinylthioacetic acid) was administered to *Ppara*^fl/fl^ and *Ppara*^ΔSMC^ mice infused with Ang II. As shown, Wy14643 significantly attenuated the Ang II-induced elevation of both systolic blood pressure (SBP) and diastolic blood pressure (DBP) in *Ppara*^fl/fl^ mice. However, this attenuation was markedly blunted in Ang II-infused *Ppara*^ΔSMC^ mice (Figure 5A,B). Echocardiogram showed that little changes of cardiac function upon Wy14643 treatment between *Ppara*^fl/fl^ and *Ppara*^ΔSMC^ mice infused with Ang II (Appendix A). Next, the vascular stiffness was assessed by PWV, PPV, and distensibility. The vascular ultrasound results showed that a Wy14643 treatment significantly improved the Ang II-induced vascular stiffness in *Ppara*^fl/fl^ mice but not in *Ppara*^ΔSMC^ mice (Figure 5C–E). The Ang II-induced increase in the intima-media thickness (IMT) was reversed by Wy14643 in *Ppara*^fl/fl^ mice but not in *Ppara*^ΔSMC^ mice (Figure 5F). These data indicate that the beneficial effect of PPARα agonist on the vascular stiffness is VSMC PPARα-dependent.

### 3.6. Wy14643 Improved Ang II-Induced Vascular Remodeling and ROS Production in a VSMC PPARα-Dependent Manner

To further test how the PPARα agonist Wy14643 protected against Ang II-induced hypertension and vascular stiffness, vascular remodeling and oxidative stress were assessed in *Ppara*^fl/fl^ and *Ppara*^ΔSMC^ mice infused with Ang II. As shown, Wy 14643 significantly inhibited the Ang II-induced vessel wall thickness (Figure 6A) and fibrosis (Figure 6B) in *Ppara*^fl/fl^ mice, whereas this inhibition effect was profoundly blunted in *Ppara*^ΔSMC^ mice. DHE staining also showed that the Ang II-primed ROS generation was markedly decreased by the Wy14643 treatment in *Ppara*^fl/fl^ mice but not in *Ppara*^ΔSMC^ mice (Figure 6C). These data indicate that the VSMC PPARα is critical for the maintenance of vascular homeostasis and involved in the protective effect of PPARα agonist on the vasculature.

## 4. Discussion

The current study demonstrates that PPARα in VSMCs plays an important role in regulating vascular remodeling and blood pressure. The chronic increase in circulating Ang II can significantly decrease the PPARα expression in VSMCs. A PPARα deficiency in VSMCs significantly exacerbates vascular remodeling upon an Ang II stimulation, leading to an increased stiffness and hypertension. Mechanistically, a PPARα deficiency augments Ang II-induced oxidative stress, especially a mitochondrial ROS accumulation in VSMCs. Moreover, the PPARα agonist Wy14643 exhibits anti-remodeling and anti-hypertension effects in a VSMC PPARα-dependent manner. Hence, PPARα represents a critical regulator of VSMC biology and vascular homeostasis.

PPARα plays a crucial role in regulating multiple vascular pathological processes, including the lipid and glucose metabolism, vascular inflammation, and atherosclerosis [33,34,35,36,37]. However, PPARα is seldom studied in hypertensive models. A PPARα agonist can lower blood pressure in a DOCA-salt-induced hypertensive mouse model by boosting renal 20-hydroxyeicosatetraenoic acid (20-HETE) synthesis and thereby reducing the sodium balance. Further, a PPARα deficiency suppresses these actions, implying a PPARα-dependence [38]. In Ang II-induced vascular injury models, the PPARα activator reduced the Ang II-induced vascular oxidative stress and inflammation [39]. In vitro studies have demonstrated that upregulating PPARα in VSMCs inhibited phenotypic switching and oxidative stress [40]. In agreement with these results, the current data revealed that *Ppara*^ΔSMC^ mice showed more adverse hypertensive vascular remodeling associated with mitochondrial oxidative damage compared to *Ppara*^fl/fl^ mice, which suggested that the endogenous protective effects of PPARα were involved in an Ang II-induced vascular injury.

Previous studies have demonstrated that the administration of the PPARα agonist Wy14643 improves the vascular function in the aorta of the spontaneously hypertensive rat [41]. Wy14643 was also shown to protect against oxidative stress and the inflammatory response evoked by a transient cerebral ischemia/reperfusion [42]. The present study revealed that the treatment of Wy14643 had a significant vascular protection in Ang II-induced mouse models, accompanied by a decreasing vascular thickness, fibrosis, and oxidative stress. However, compared to *Ppara*^fl/fl^ mice, the treatment of Wy14643 to *Ppara*^ΔSMC^ mice resulted in a more hypertensive vascular remodeling and oxidative damage. This study further revealed that the endogenous PPARα of VSMCs protects against Ang II-induced hypertensive vascular remodeling, and Wy14643 improves hypertensive vascular remodeling partially through VSMC-PPARα. Future studies are needed to investigate whether the PPARα–mitochondrial oxidative damage mechanisms account for other vascular cell types in a vascular injury. Meanwhile, Ang II elevates the systemic ROS levels and mitochondrial oxidative stress, which have both been implicated in the pathology of a vascular dysfunction [43]. Vascular mitochondria have emerged as an important player in maintaining vascular homeostasis. As such, age- and disease-related impairments in the mitochondrial function contribute to vascular stiffness [44]. The PPARα agonist fenofibrate has been shown to improve the vascular stiffness in obesity patients [45]. In line with these studies, *Ppara*^ΔSMC^ mice showed more mitochondrial ROS production and a more pronounced reduction in the mitochondrial membrane potential in Ang II-treated VSMCs (Figure 4). It is worth exploring the mechanistic link between PPARα and the mitochondrial function in VSMCs on a more detailed molecular basis. In addition to its role in attenuating the mitochondrial ROS production and maintaining the mitochondrial membrane potential, our previous work has also demonstrated that a PPARα deficiency can attenuate VSMC apoptosis induced by Ang II and hydrogen peroxide, and increase the migration of Ang II-challenged VSMCs [22], thus indicating a profound and complex role for PPARα in maintaining VSMCs homeostasis. Although we detected a PPARα reduction in VSMCs accelerated Ang II-induced mitochondrial oxidative stress, we did not further explore the PPARα target genes in VSMCs to establish a direct mechanistic link between mitochondrial dysfunction and hypertensive vascular remodeling. Although we observed that a PPARα activation by Wy14643 attenuated vascular remodeling and hypertension in a PPARα deficiency, it is noteworthy that Wy14643 only partially through VSMC-PPARα improves hypertensive vascular remodeling. Future studies are needed to investigate whether the PPARα-modulated and mitochondrial oxidative damage-related mechanisms account for the pathophysiology of other vascular cell types in a vascular injury.

## 5. Conclusions

In summary, we showed that a PPARα deficiency in VSMCs significantly exacerbated the hypertension, vascular remodeling, stiffness, and oxidative stress. In addition, the effect of PPARα is partially mediated by decreasing mitochondrial oxidative stress, suggesting that VSMC-PPARα is protective against vascular dysfunction in conditions with an Ang II induction.

## Figures and Tables

**Figure 1 antioxidants-11-02378-f001:**
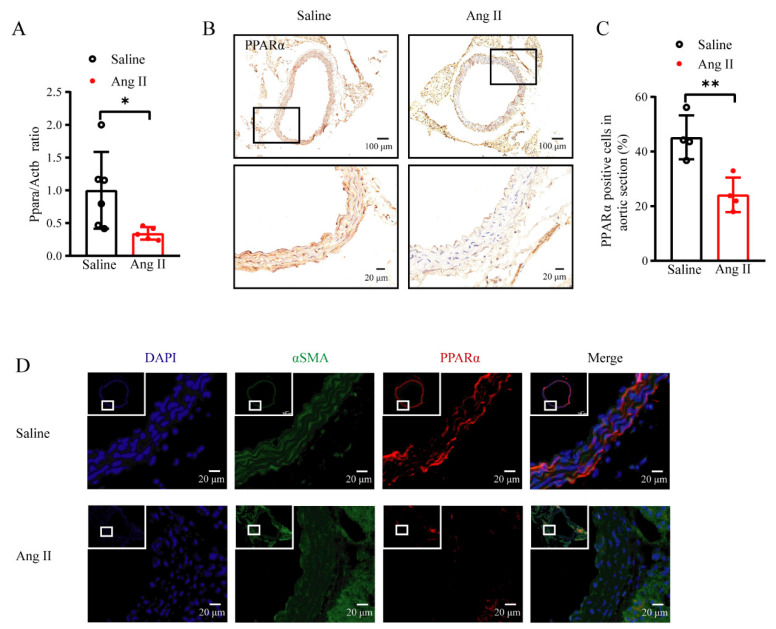
*Ppara* was decreased in Ang II−induced vascular remodeling. (**A**) mRNA levels of *Ppara* in aortas of Ang II infusion WT mice compared with that of saline infusion WT mice (*n* = 5–6, unpaired *t*-test, * *p* < 0.05). (**B**) Immunohistochemistry staining for PPARα in aortas of Ang II infusion mice and saline infusion mice. (**C**) Immunohistochemistry analysis of PPARα protein level (*n* = 4, unpaired *t*-test, * *p* < 0.05, ** *p* < 0.01). (**D**) Immunofluorescence staining for PPARα in aortas of Ang II infusion mice and saline infusion mice. (*n* = 3, blue represents DAPI staining, green represents αSMA staining and red represents PPARα staining). Data are means ± SEM. * *p* < 0.05, ** *p* < 0.01 between groups.

**Figure 2 antioxidants-11-02378-f002:**
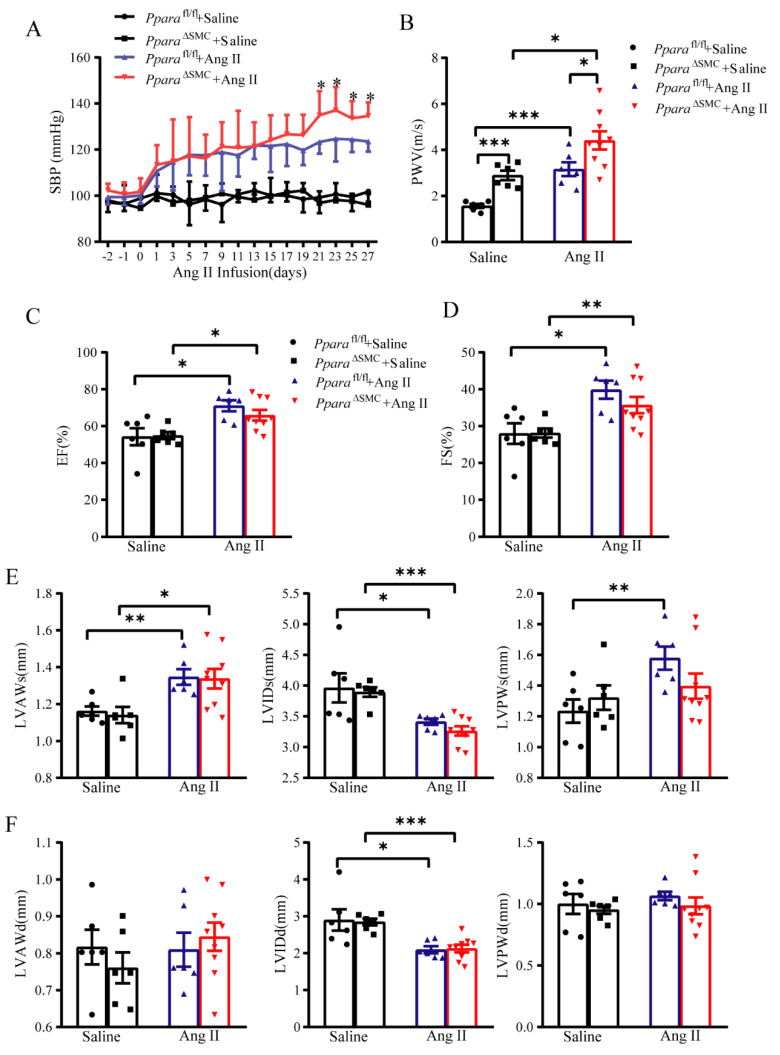
PPARα deficiency in VSMCs aggravated Ang II−induced hypertension and vascular stiffness independent of cardiac function. (**A**) Systolic blood pressure (SBP) was measured by noninvasive tail–cuff method before and after saline or Ang II infusion (490 ng/kg/day). (**B**) Pulse wave velocity (PWV) was measured by vascular ultrasound after Saline or Ang II infusion. (**C**) Ejection fraction (EF) and (**D**) fractional shortening (FS) were assessed by two-dimensional echocardiography after saline or Ang II infusion. Left ventricular anterior wall (LVAW) thickness, left ventricular internal diameter (LVID), and left ventricular posterior wall (LVPW) thickness at end-systole (s, **E**) or end-diastole (d, **F**) are recorded and analyzed (*n* = 6–9, two-way ANOVA). Data are means ± SEM. * *p* < 0.05, ** *p* < 0.01, *** *p* < 0.01 between groups.

**Figure 3 antioxidants-11-02378-f003:**
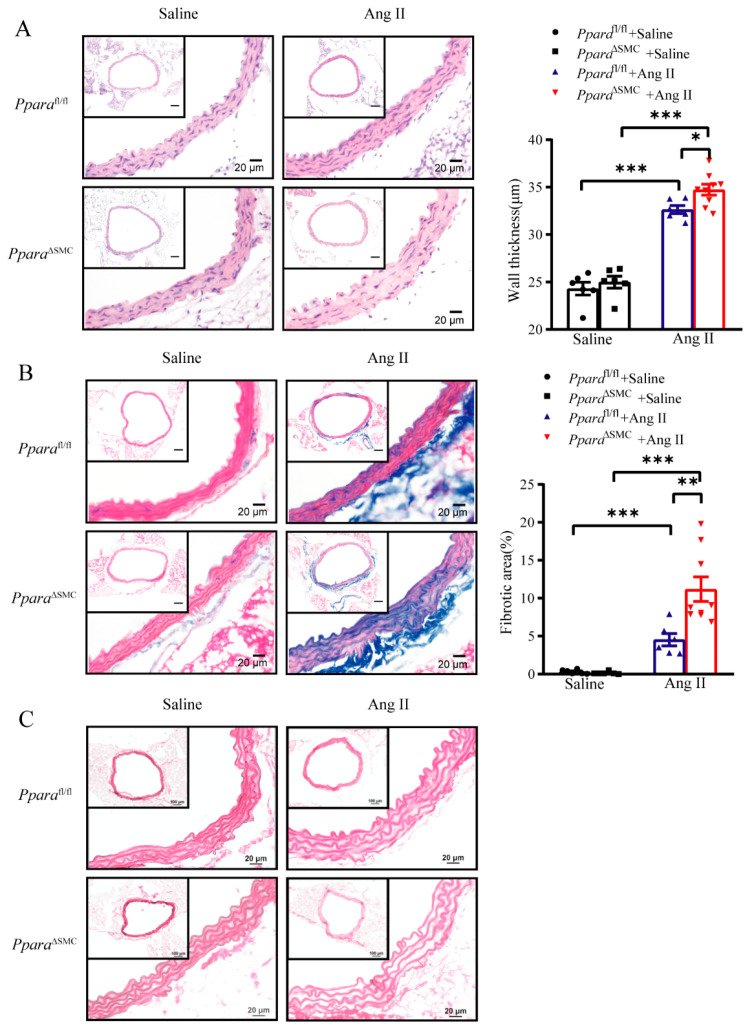
PPARα deficiency in VSMCs aggravated Ang II−induced vascular remodeling (**A**) Representative images of hematoxylin and eosin-stained aortic cross sections are shown, aortic thickness was quantified. (**B**) Representative images of Masson’s trichrome-stained aortic cross sections are shown, and fibrotic area was quantified. (**C**) Representative images of Weigert’s elastin -stained aortic cross sections are shown (*n* = 6–9, two-way ANOVA). Data are means ± SEM. * *p* < 0.05, ** *p* < 0.01, *** *p* < 0.01 between groups.

**Figure 4 antioxidants-11-02378-f004:**
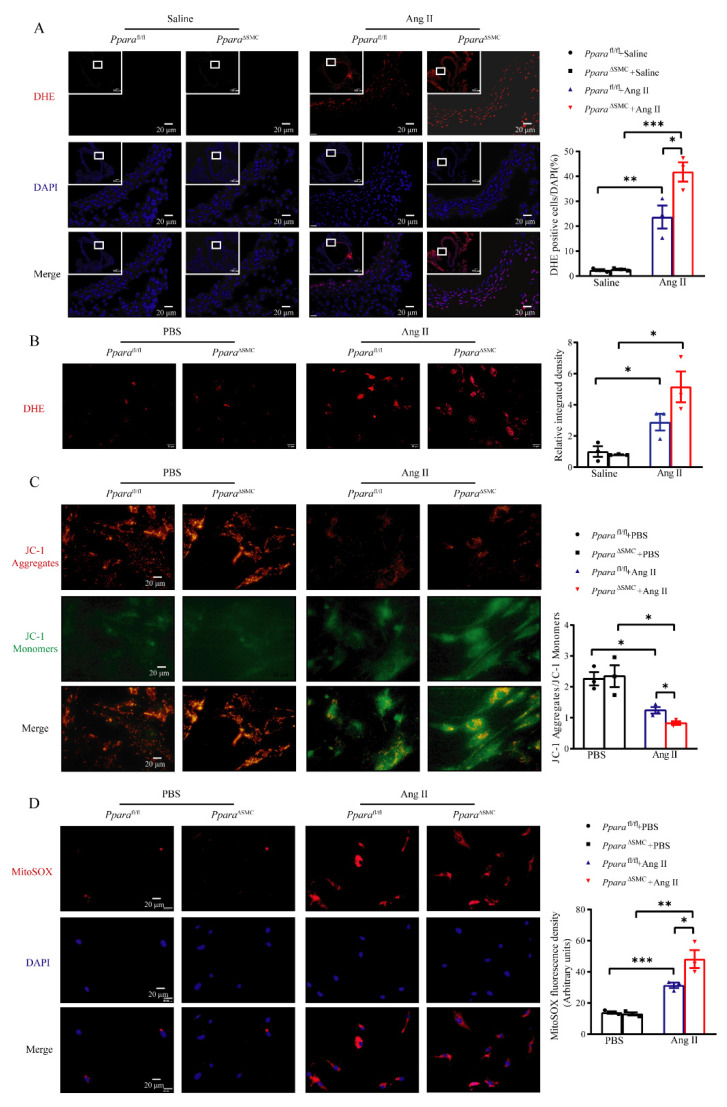
PPARα deficiency in VSMCs aggravated Ang II−induced vascular oxidative stress (**A**) Representative images and analysis of dihydroethidium (DHE) staining for aortic cross sections from *Ppara*^ΔSMC^ and *Ppara*^fl/fl^ mice. Scale bar = 20 μm. (**B**) Representative images and analysis of DHE staining for VSMCs from *Ppara*^ΔSMC^ and *Ppara*^fl/fl^ mice. Scale bar = 20 μm. (**C**) Representative images and analysis of JC-1 aggregates and JC-1 monomers in VSMCs from *Ppara*^ΔSMC^ and *Ppara*^fl/fl^ mice were treated with Ang II (1 μmol/L) for 24 h. The orange fluorescence represents the superposition of green fluorescence (JC-1 monomer) and red fluorescence (JC-1 aggregate). Scale bar = 20 μm. (**D**) Representative images and analysis of MitoSOX staining for VSMCs from *Ppara*^ΔSMC^ and *Ppara*^fl/fl^ mice were treated with Ang II (1 μmol/L) for 24 h (scale bar = 20 μm, *n* = 3, two-way ANOVA). Data are means ± SEM. * *p* < 0.05, ** *p* < 0.01, *** *p* < 0.01 between groups.

**Figure 5 antioxidants-11-02378-f005:**
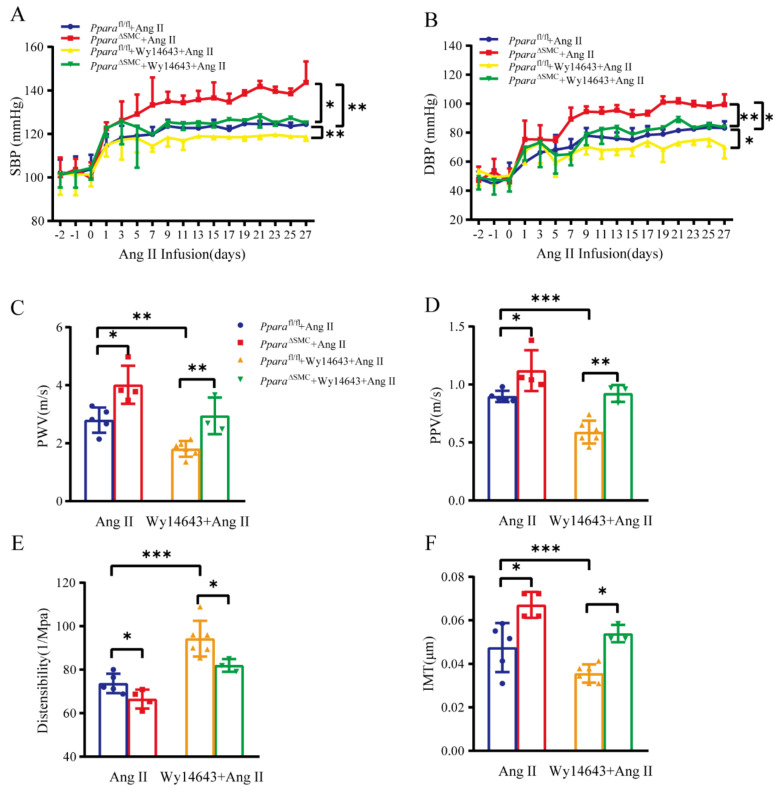
Activation of PPARα by Wy14643 attenuated Ang II−induced hypertension and vascular stiffness. (**A**) Systolic blood pressure (SBP) and (**B**) diastolic blood pressure (DBP) was measured by noninvasive tail–cuff method before and after Wy14643 treatment in Ang II infusion mice (490 ng/kg/day). (**C**) Pulse wave velocity (PWV), (**D**) pulse pressure variation (PPV), (**E**) distensibility, and (**F**) intima-media thickness (IMT) were measured by vascular ultrasound (*n* = 3–6, two-way ANOVA). Data are means ± SEM. * *p* < 0.05, ** *p* < 0.01, *** *p* < 0.01 between groups.

**Figure 6 antioxidants-11-02378-f006:**
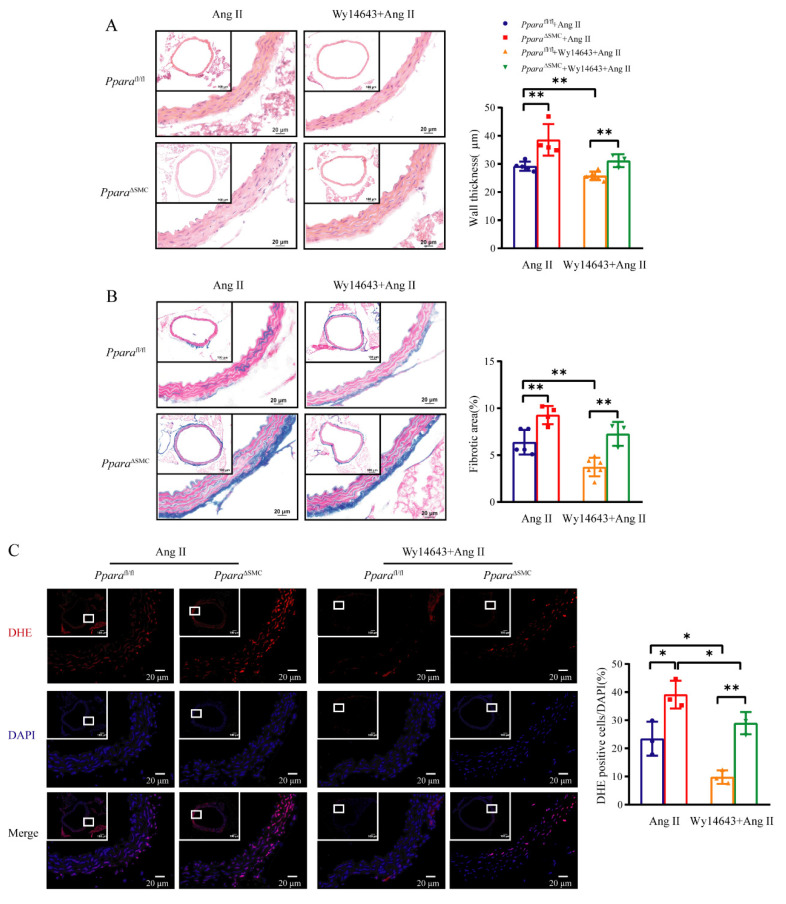
Activation of PPARαby Wy14643 improved Ang II−induced vascular remodeling and ROS production. Representative images and analysis of hematoxylin and eosin (H&E) staining (**A**), Masson’s trichrome staining (**B**), and dihydroethidium (DHE) staining (**C**) for aortic cross sections from *Ppara*^ΔSMC^ and *Ppara*^fl/fl^ mice with Wy14643 feeding and Ang II infusion. Scale bar = 20 μm (*n* = 3–6, two-way ANOVA). Data are means ± SEM. * *p* < 0.05, ** *p* < 0.01 between groups.

## Data Availability

The data presented in this study are available in this manuscript.

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
