# Peer review of "Peroxisome Proliferator-Activated Receptor α Attenuates Hypertensive Vascular Remodeling by Protecting Vascular Smooth Muscle Cells from Angiotensin II-Induced ROS Production"

_antioxidants, 2022, doi:10.3390/antiox11122378_

Round 1

Reviewer 1 Report

Ye Liu and collaborators investigated the impact of PPARα deficiency in the remodelling of VSMC induced by angiotensin II and in hypertension development. This work is well-conducted, interesting and well-written. Nevertheless, several controls and additional experiments lack and must be integrated to clearly support their conclusion.

1/ About PpparαΔSMC mouse model and analysed parameters. Authors use PpparαΔSMC mice which are not sufficiently referenced and/or characterized to appreciate the results displayed in this work:

-        * Is this model the same used in Duan Y et al BBR 2021? The authors must correctly reference their model or at least shown the expression of PPARα in the tissues of interest.

-          * What is the age of the mice at the time of experiments?

-         * What are the metabolic general parameters of mice in all conditions (body and fat weight, food and drink intake, blood parameters in term of lipid and carbohydrates….)? This is required to eliminate any accessory effects which could mask/disturb some part of the analysis.

-         *  Why different parameters were analysed between Fig 2 and 5? In addition, authors must add control group in Fig 5 and 6 (saline +/- Wy in fl/fl and ΔSMC mice), which will fix a part of the discrepancies between Fig 2 and 5.

2/ About primary VSMC experiments:

-        *   DHE staining could be added in Fig 4 to correlate in vivo and in vitro experiments.

-         *  Is it disturbing that the authors have not added rescue experiments by adding Wy14643 in all analysis. This should be added.

-        *  Authors write about oxidative stress but analysed superoxide anion production in vitro and in vivo by using the specific DHE and MitoSOX probes. Even if their results shown interesting modulation of ROS production this does not represent “oxidative stress”, as ROS can be cleared by antioxidant defence. If the authors want to prove the presence of oxidative stress, they must use more adequate probe to assay hydroxyl radical (CMH2DCFDA), peroxynitrite (APF or HPF) or oxidative stress damage, as lipid peroxidation or protein carbonylation. This is a crucial point impacting all the manuscript from the title to the conclusion.

3/ The authors claimed several times that PPARa deficiency accelerated the effect of Angiotensin, but there is no time point analysis in this work. I think that the term “accelerate” should be replace by “aggravate” or other.

4/ A comparison with previous authors’ work (Duan BBR 2021) could be added in the discussion to carry additional data about the impact of PPARa deficiency in VSMC.

Author Response

Reviewer 1

Comments and Suggestions for Authors

Ye Liu and collaborators investigated the impact of PPARα deficiency in the remodeling of VSMC induced by angiotensin II and in hypertension development. This work is well-conducted, interesting and well-written. Nevertheless, several controls and additional experiments lack and must be integrated to clearly support their conclusion.

Response: We thank the reviewer for the positive comments. According to reviewer’s suggestions, we have performed additional experiments to further support our findings and revised the manuscript accordingly. Please refer to the responses as below.

Comment 1. About PpparαΔSMC mouse model and analysed parameters. Authors use PpparαΔSMC mice which are not sufficiently referenced and/or characterized to appreciate the results displayed in this work:

Comment 1.1 Is this model the same used in Duan Y et al BBR 2021? The authors must correctly reference their model or at least shown the expression of PPARα in the tissues of interest.

Response:We thank the reviewer for the key concerns. Yes, the same VSMC-specific PPARa-deficient mouse model as described in Duan Y et al BBR 2021 were used in this study. We now cited this paper in Materials and Methods section (page 2) in the revised manuscript.

Comment 1.2 What is the age of the mice at the time of experiments?

Response:Two-month-old male Pparafl/fl and PparaΔSMC mice were infused with saline or Ang II, which was now addressed in Materials and Methods in 2.1 in Page 2.

Comment 1.3 What are the metabolic general parameters of mice in all conditions (body and fat weight, food and drink intake, blood parameters in term of lipid and carbohydrates….)? This is required to eliminate any accessory effects which could mask/disturb some part of the analysis.

Response:We thank the reviewer for the critical concerns. No significant changes of metabolic general parameters were found between Pparafl/fl and PparaΔSMC mice, including body and fat weight, food and drink intake, fasting glucose, serum triglycerides and cholesterol. These data were now listed in the supplementary Fig1 in the revised version as below.

Supplementary Figure 1. The metabolic general parameters of PparaΔSMC and Pparafl/fl mice. (A) Body Weight, eWAT(white adipose tissue), and BAT(brown adipose tissue), (B)Food intake, drink intake ,and fasting blood-glucose in PparaΔSMC and Pparafl/fl mice.  (C) Triglyceride, total cholesterol, and free cholesterol in plasma of PparaΔSMC and Pparafl/fl mice. (n=6)

Comment 1.4 Why different parameters were analysed between Fig 2 and 5? In addition, authors must add control group in Fig 5 and 6 (saline +/- Wy in fl/fl and ΔSMC mice), which will fix a part of the discrepancies between Fig 2 and 5.

Response:We thank the reviewer for the important comment and suggestion. We first confirmed that VSMC PPARα deficiency in VSMCs exacerbated Ang II-induced vascular stiffness as shown in Fig 2, however, no significant differences in cardiac function were found when compared to Pparafl/fl mice. We now included these data in supplemental Fig 2 in the revised manuscript as below. Consequently, we only focused on the vascular but not cardiac parameters in Fig 5.

We totally agree with the reviewer that it would be perfect to include control mice (saline +/- Wy in Pparafl/fl and PparaΔSMC mice) in Fig 5 and 6. Previous studies reported that Wy14643 did not affect vascular function in basal conditions[1]. COVID-19 had strong impact on animal clonal expansion. We applied all the mice into Ang II-treated group due to the limited animal amount. We thank the reviewer for the kindly understanding.

Supplementary Figure 2. Activation of PPARα by Wy14643 had no effect on Ang II-induced cardiac dysfunction. (A) EF and FS were assessed by two-dimensional echocardiography after Wy14643 diet. (B) LVAWs and LVAWd. (C) LVIDs and LVIDd, (D) LVPWs and LVPWd are recorded and analyzed (n = 3-6).

  1. About primary VSMC experiments:

Comment 2.1 DHE staining could be added in Fig 4 to correlate in vivo and in vitro experiments.

Response:We thank the reviewer for this constructional suggestion. We have added DHE staining in vitro in Fig4B in the revised version as below.

(B) Representative images and analysis of DHE staining for VSMCs from PparaΔSMC and Pparafl/fl mice.  (Scale bar= 20 μm, n = 3, two-way ANOVA). Data are means ± SEM. * p < 0.05 between groups.

Comment 2.2 Is it disturbing that the authors have not added rescue experiments by adding Wy14643 in all analysis. This should be added.

Response:We appreciate the reviewer for the key concerns. We performed the rescue experiments directly in vivo as shown in Fig 5 and Fig 6, which we thought was more convincing than cellular experiments in vitro. Hence, we did not include the rescue experiments in cells in vitro.

Comment 2.3 Authors write about oxidative stress but analysed superoxide anion production in vitro and in vivo by using the specific DHE and MitoSOX probes. Even if their results shown interesting modulation of ROS production this does not represent “oxidative stress”, as ROS can be cleared by antioxidant defence. If the authors want to prove the presence of oxidative stress, they must use more adequate probe to assay hydroxyl radical (CMH2DCFDA), peroxynitrite (APF or HPF) or oxidative stress damage, as lipid peroxidation or protein carbonylation. This is a crucial point impacting all the manuscript from the title to the conclusion.

Response: We totally agree with the reviewer that the modulation of ROS production does not represent “oxidative stress”. Although we believed that exacerbated oxidative stress exited in Ang II-treated PparaΔSMC mice, we did not show sufficient evidence due to time limit. We have softened the expression from oxidative stress to ROS production throughout the manuscript as suggested.

Comment 3. there is no time point analysis in this work. I think that the term “accelerate” should be replace by “aggravate” or other.

Response:We have replaced the term of “accelerate” by “aggravate” or “exacerbate” as suggested.

Comment 4. A comparison with previous authors’ work (Duan BBR 2021) could be added in the discussion to carry additional data about the impact of PPARa deficiency in VSMC.

Response:We appreciate the reviewer for pointing out the citation and comparison problem. We have incorporated additional discussion in the revised manuscript (page 13,14) to provide more comprehensive information as to the impact of PPARα deficiency on VSMC pathophysiology.  

Besides its role in attenuating mitochondrial ROS production and maintaining mitochondrial membrane potential, our previous work has also demonstrated that PPARα deficiency can attenuate VSMC apoptosis induced by Ang II and hydrogen peroxide, and increase the migration of Ang II-challenged VSMCs[2], thus indicating a profound and complex role for PPARα in maintaining VSMCs homeostasis.

Reviewer 2 Report

Please see attached

Author Response

Reviewer 2

Comments and Suggestions for Authors

The article is well written and shows very nice pictures of the histological sections of the vessels as well as providing important information to the literature.

I am in favor of publication and suggest some changes and ask some questions for you.

 Paragraph 2.4. Histological analysis Mice were sacrificed by CO2 inhalation and perfused with saline. Aortas from Pparafl/fl and PparaSMC mice were fixed in phosphate-buffered 4% formalin for 24 hours and then embedded in Tissue-Tek O.C.T. compound (4583, Sakura Finetek USA, CA, USA). Frozen sections (7 μm) of the thoracic aorta from Pparafl/fl and PparaSMC mice were continuously collected from the proximal to the distal. Hematoxylin (RY-ICH001a, Roby, Beijing, 117 China) and eosin (RY-ICH002a, Roby, Beijing, China) staining, elastin staining by Weigher’s elastic staining method (G1592, Solarbio, Beijing, China), and Masson’s tri- chrome staining were routinely performed as previously described[25]. Images were obtained using an Olympus laser scanning microscope (U-LH100-3, Olympus Corporation, 121 TKY, JPN).

Comment 1. Why did you fix in formalin and didn't directly use OCT for Frozen ?????

Response: Thank you for your comments.  Admittedly, the use of frozen tissue sections has advantages including the retention of the natural protein structure of antigens and the avoidance of the usage of formalin, which is toxic and carcinogenic. However, it cannot avoid the disadvantages, including tissue morphology loss due to the freezing artifact and the thickness of cryostat-generated sections. Formalin-fixed paraffin-embedded (FFPE) tissue section is recognized as the most commonly used method of tissue preparation for diagnostic histopathology[3], which can avoid the above-mentioned drawbacks of frozen tissue sections and yield excellent tissue morphology. Taken together with the ease of integration into routine tissue processing and the rapidly expanding number of commercially available antibodies that have been demonstrated to react with formalin-fixed tissues, we selected FFPE tissues for use in applications in the present study.

Comment 2. Have you used cryostat sections for Masson and Weigher ???

Response: Thank you for your comments. Cryostat sections were used for Masson’s trichrome (Sigma-Aldrich, HT15-1KT) and Weigher’s staining (G1592, Solarbio, Beijing, China) according to the manufacturer’s instructions.

Comment 3. (H&E) Hematoxylin and eosin,Did you also observe the H&E sections with Olympus laser scanning microscope ???? why????? The images that you report below in the text are in optical microscopy.

Response: We apologize for the unclear writing. Olympus BX53 has both optical and fluorescent modules. H&E images were obtained under the optical mode. We now corrected the statement into Olympus microscopy ((BX53, Olympus Corporation, 121 TKY, JPN) in the revised manuscript.

Comment 4. Make sure that throughout the text Masson and Weigher are spelled correctly.

Response: We have checked all the spelling mistakes throughout the text as suggested.

Comment 5. Paragraph 2.5. Immunohistochemistry staining Frozen sections?

Response: Yes, immunohistochemistry staining was performed on frozen sections.

General comments: The H&E, histochemical and fluorescence figures are very beautiful. Check the captions and if you can, provide a better resolution.

Good work.

Response: Thank you for your positive comments. According to your suggestions, we have corrected the mistakes and revised the manuscript accordingly. We would like very much to follow the reviewer’s suggestion and provide images with the best possible quality. However, the resolution of the images is the highest available obtained on the image-acquisition apparatus, and thanks again for the reviewer’s favorable appreciation of our work.

Reviewer 3 Report

1. Could you give the composition of the chow diet in the supplementary or add a reference to the chow diet?

2. Was Wy diet commenced two weeks after initiating the AngII infusion?

3. WY-14643Pirinixic acidenhances both PPARα and PPARγ activity, but only PPARα is mentioned in the manuscript. Do you think that PPARγ may play a role in attenuating Ang II-induced hypertension and vascular stiffness? You should assess the induction of PPARα- and PPARγ-regulated genes.

4.  PPARα deficiency aggravated Ang II-induced ROS in VSMCs. Additionally, Wy reduced Ang II-induced ROS in a PPARα-dependent manner. What are the mechanisms? You should assay the expression of ROS-related genes.  

5. Could you specify the N for each group in the legend of all figures and Why the N of some groups are only 3? This is statistically weak.

6. Some staining quantification results did not match the representative images. Please replace with an appropriate representative image.

7. In Fig.3A, the quantification results showed the aortic thickness of [Ppara ΔSMC+saline] is thicker than that in [Pparafl/fl+saline], but in the image, [Pparafl/fl+saline] is thicker.

8. In Fig.4C, the quantification results showed the MitoSOX fluorescence density of [Ppara ΔSMC+PBS] is slightly lower than that in [Pparafl/fl+PBS], but in the image, [Pparafl/fl+PBS] is lower.

9. There are three uppercase ANDs on lines 272, 274, 277.

10 There are some grammatical errors. For example, line 49, hemodynamics. Line 94, you need to show full term of PWV, etc. Please check your manuscript more carefully after resubmission.

Author Response

Reviewer 3

Comments and Suggestions for Authors

Comment 1. Could you give the composition of the chow diet in the supplementary or add a reference to the chow diet?

Response: We thank the reviewer for the important concern. The standard chow diet was supplied by the animal core facility. The diet was composed of 18% from protein, 4% from fat, 5% from crude fibr, 8% from coarse ash, 10% from water, 0.82% from Lysine, 1.0-1.8% from Calcium, 0.6-1.2% from Phosphorus, and 0.3-0.8% from salt, which were listed in Supplementary Table 1.

Comment 2. Was Wy diet commenced two weeks after initiating the AngII infusion?

Response: We apologize for the unclear writing. The Wy14643 diet was initiated from day 0 upon Ang II infusion, which was now clearly stated in the Materials and Methods section.

Comment 3. WY-14643(Pirinixic acid)enhances both PPARα and PPARγ activity, but only PPARα is mentioned in the manuscript. Do you think that PPARγ may play a role in attenuating Ang II-induced hypertension and vascular stiffness? You should assess the induction of PPARα- and PPARγ-regulated genes.

Response: We thank the reviewer for the critical concern. However, we politely disagree with the reviewer that Wy-14643 enhanced both PPARa and PPARg activity. Numerous studies have demonstrated that WY-14643 is a potent PPARα but not PPARg agonist [4-5].

Using VSMC PPARa-deficient mice, we demonstrated that the protective role of PPARa agonist Wy-14643 is mainly, or at least more than half, VSMC PPARa dependent. Of course, we did not exclude the possibly protective role of VSMC PPARg in Ang II-induced vascular dysfunction. In fact, Y Eugene Chen’s group published paper that PPARg plays an important role in VSMC biology and vascular homeostasis. Consistently, we found that both PPARa and PPARg was downregulated in VSMCs upon vascular injury (unpublished data in another story). We totally understand the reviewer’s concerns and hope to clearly address it in the future study.

Comment 4.  PPARα deficiency aggravated Ang II-induced ROS in VSMCs. Additionally, Wy reduced Ang II-induced ROS in a PPARα-dependent manner. What are the mechanisms? You should assay the expression of ROS-related genes.  

Response: We thank the reviewer for the constructive suggestion, which was of great help to improve our manuscript. For the mechanism of increased ROS production in Ang II-treated PPARa-deficient VSMCs, we supposed that mitochondrial OXPHOS was impaired under PPARa deficiency. We measured the mRNA level of NOX4, a main isoform of NADPH oxidase (NOX) expressed in VSMCs [6]. Ang II induced a slight increase of Nox4 mRNA levels in Pparafl/fl mice, however, PPARa deficiency dramatically augmented this effect. We also realized that determination of mitochondrial function would replenish the mechanistic understanding. We hope to perform these analyses in the future study. Currently, we included the Nox4 mRNA data in the revised manuscript.

NADPH oxidase (NOX) family is a major source of vascular oxidative stress and NOX4 is the main isoform expressed in VSMCs of large arteries [3]. Our further investigation showed that Ang II-induced the vascular elevation of Nox4 in Pparafl/fl mice. However, this effect was slightly increased by PPARα deletion in PparaΔSMC mice (Supplementary Figure 3). NOX4 is recognized as an oncoprotein localized to mitochondria[7]. The expression of NOX4 has been shown to be induced in VSMC mitochondria and vasculature and is correlated with increased aortic stiffening and atherosclerosis in aged mice[8].

Supplementary Figure 3. PPARα deficiency in VSMCs aggravated Ang II-induced the vascular elevation of Nox4 (A) The mRNA level of Ppara and Nox4 in VSMCs from PparaΔSMC and Pparafl/fl mice were treated with Ang II (1 μmol/L) for 24 hours. (n = 3, two-way ANOVA). Data are means ± SEM. * p < 0.05between groups.

Comment 5. Could you specify the N for each group in the legend of all figures and Why the N of some groups are only 3? This is statistically weak.

Response: As suggested, the number of each group in the legend has been labelled in the figure legends. We also noted that some of the sample number was small. This was mainly due to COVID19-caused colony cutdown.

Comment 6. Some staining quantification results did not match the representative images. Please replace with an appropriate representative image.

Response: We have checked all the morphological images and replaced with representative ones as reviewer suggested.

Comment 7. In Fig.3A, the quantification results showed the aortic thickness of [Ppara ΔSMC+saline] is thicker than that in [Pparafl/fl+saline], but in the image, [Pparafl/fl+saline] is thicker.

Response: We appreciate the reviewer for pointing out this problem. We now replaced them with representative images as shown below.

Fig. 3A

Comment 8. In Fig.4C, the quantification results showed the MitoSOX fluorescence density of [Ppara ΔSMC+PBS] is slightly lower than that in [Pparafl/fl+PBS], but in the image, [Pparafl/fl+PBS] is lower.

Response: We now replaced them with representative images as shown below.

Fig. 4C

Comment 9. There are three uppercase ANDs on lines 272, 274, 277.

Response: We apologize for the mistake. We have corrected them as indicated.

Comment 10. There are some grammatical errors. For example, line 49, hemodynamics. Line 94, you need to show full term of PWV, etc. Please check your manuscript more carefully after resubmission

Response: We sincerely appreciate the reviewer for the careful reading of our manuscript. We now checked the text thoroughly and corrected the grammatical errors.

Round 2

Reviewer 1 Report

The authors' responses significantly improved the quality of the manuscript and the robustness of the conclusion. The manuscript can be published in the present form.

Reviewer 3 Report

Please correct line 270 and 271, PparaF/F etc. before the publication.